# MicroRNA Inhibiting Atheroprotective Proteins in Patients with Unstable Angina Comparing to Chronic Coronary Syndrome

**DOI:** 10.3390/ijms251910621

**Published:** 2024-10-02

**Authors:** Michał Kowara, Michał Kopka, Karolina Kopka, Renata Głowczyńska, Karolina Mitrzak, Dan-ae Kim, Karol Artur Sadowski, Agnieszka Cudnoch-Jędrzejewska

**Affiliations:** 1Chair and Department of Experimental and Clinical Physiology, Laboratory of Centre for Preclinical Research, Medical University of Warsaw, 1b Banacha Street, 02-097 Warsaw, Polandagnieszka.cudnoch-jedrzejewska@wum.edu.pl (A.C.-J.); 2Department of Methodology, Laboratory of Centre for Preclinical Research, Medical University of Warsaw, 1b Banacha Street, 02-097 Warsaw, Poland; 31st Department of Cardiology, Medical University of Warsaw, 1a Banacha Street, 02-097 Warsaw, Poland

**Keywords:** atherosclerosis, stable plaque, vulnerable plaque, unstable angina, chronic coronary syndrome, microRNA, atheroprotective factors

## Abstract

Patients with unstable angina present clinical characteristics of atherosclerotic plaque vulnerability, contrary to chronic coronary syndrome patients. The process of athersclerotic plaque destabilization is also regulated by microRNA particles. In this study, the investigation on expression levels of microRNAs inhibiting the expression of proteins that protect from atherosclerotic plaque progression (miR-92a inhibiting KLF2, miR-10b inhibiting KLF4, miR-126 inhibiting MerTK, miR-98 inhibiting IL-10, miR-29b inhibiting TGFβ1) was undertaken. A number of 62 individuals were enrolled—unstable angina (UA, *n* = 14), chronic coronary syndrome (CCS, *n* = 38), and healthy volunteers (HV, *n* = 10). Plasma samples were taken, and microRNAs expression levels were assessed by qRT-PCR. **As a result, the** UA patients presented significantly increased miR-10b levels compared to CCS patients (0.097 vs. 0.058, *p* = 0.033). Moreover, in additional analysis when UA patients were grouped together with stable patients with significant plaque in left main or proximal left anterior descending (“UA and LM/proxLAD” group, *n* = 29 patients) and compared to CCS patients with atherosclerotic lesions in other regions of coronary circulation (“CCS other” group, *n* = 25 patients) the expression levels of both miR-10b (0.104 vs. 0.046; *p* = 0.0032) and miR-92a (92.64 vs. 54.74; *p* = 0.0129) were significantly elevated. **In conclusion**, the study revealed significantly increased expression levels of miR-10b and miR-92a, a regulator of endothelial protective KLF factors (KLF4 and KLF2, respectively) in patients with more vulnerable plaque phenotypes.

## 1. Introduction

One of the most prevalent diseases worldwide is ischemic heart disease, which affects 126 million individuals (1.665 per 100,000), according to the Global Burden of Disease (GBD) Registry [1]. This disease is presented either in a stable phenotype and called chronic coronary syndrome or in unstable phenotype and called acute coronary syndrome [2,3]. Acute coronary syndrome can be manifested as two diseases. One disease is unstable angina, which is characterized by new clinical symptoms (mainly chest pain) due to ischemia derived from a sudden coronary artery narrowing by a thrombus on a ruptured plaque. However, when the duration or character of ischemia is sufficient to cause myocardial tissue necrosis, myocardial infarction develops [4]. From a pathophysiological point of view, chronic coronary syndrome is a clinical manifestation of stable atherosclerotic plaque, which narrows a coronary artery and causes insufficient blood flow and insufficient oxygen supply during increased oxygen demand (e.g., during physical exercise of a certain intensity) [5]. Importantly, a stable coronary plaque is composed mainly of lipid-laden macrophages called “foam cells” surrounded by an extracellular matrix and covered by a thick fibrous cap, which makes the entire structure resistant to rupture. In contrast, unstable coronary plaque is composed of foam cells, lipid droplets, a significant number of inflammatory cells, and necrotic fields surrounded by fragmented extracellular matrix and covered by a thin fibrous cap, which makes the plauqe structure vulnerable to rupture [6,7]. The rupture leads to contact between blood and extracellular matrix, subsequent thrombosis and sudden coronary artery narrowing which makes the coronary artery incapable of maintaining appropriate oxygenation to the culprit myocardium [8]. The process of atherosclerotic plaque progression and destabilization depends on many molecular pathways, especially proinflammatory. According to Silvestre-Roig et al., necrotic core formation, inflammatory cell infiltration into the plaque, and extracellular matrix remodeling are principal components of atherosclerotic plaque destabilization [9]. Moreover, these processes are enhanced by proinflammatory-activated endothelial cells [10]. However, the processes that protect from atherogenesis also exist within the plaque. The atheroprotective factors targeting the aforementioned components of plaque destabilization are IL10 (an anti-inflammatory cytokine that inhibits infiltrating immune cells), MerTK (a factor that inhibits efferocytosis, i.e., plaque clearance by macrophages and prevents necrotic cores generation) and TGFβ (especially TGFβ1 that promotes stable phenotype of the plaque through extracellular matrix remodeling) [11,12,13]. Due to the crucial role of the endothelium in the process of atherogenesis, factors that promote the atheroprotective phenotype of endothelial cells, i.e., KLF2 and KLF4, are also of utmost importance. Noteworthily, all these processes are under the control of epigenetic regulators [14,15,16,17]. One type of such regulators are small non-coding RNA particles called microRNAs, whose main function is to bind to complementary mRNA and cause its degradation through the RICS complex [18]. MicroRNAs are considered regulatory factors that interfere with a huge network of different physiological and pathophysiological processes, including atherogenesis [19]. Apart from that, microRNAs are relatively stable particles, and because of this, they are also considered potential biomarkers of pathophysiological processes (like atherogenesis) [20]. Many studies were undertaken on microRNAs targeting pro-atherogenic factors [21]. In this paper, however, we aimed to find microRNA that inhibits aforementioned atheroprotective factors (IL10, MerTK, TGFβ1, KLF2, and KLF4) and assess their expression levels in patients with unstable angina compared to patients with chronic coronary syndrome and healthy volunteers.

## 2. Results

### 2.1. Baseline Characteristics

Healthy volunteers were younger than patients from the other groups. Unstable angina patients presented higher BMI, lower HDL cholesterol, and higher AlAT (alanine aminotransferase) levels. There were also more active smokers in the UA population. In addition to the younger age, healthy volunteers presented with higher eGFR, which reflects better kidney filtration. All baseline characteristics are presented in Table 1. 

### 2.2. MicroRNA Expression Levels—The Main Comparison between Study Groups

The expression levels of miR-92a, miR-10b, miR-126. miR-98 and miR-29b in all three study groups, and statistical significance values (*p*) are presented in Table 2 and Figure 1. The reciprocal analysis of miR-10b expression levels in all groups revealed a significant difference in miR-10b expression levels in UA and CCS, which is presented in Figure 2. There were no other significant differences between other groups and other microRNAs. 

### 2.3. MicroRNA Expression Levels—Additional Comparison: Only Men Considered

According to the baseline characteristics, there were more men in UA group than in CCS group and HV group. Therefore, all study microRNAs were also compared only in men (women excluded) and only for CCS and UA groups (due to a low number of male healthy volunteers, i.e., n = 4, the HV group was excluded from this study), and the results were presented in Table 3 and Figure 3. It was revealed that male UA patients presented significantly more elevated miR-10b levels than male CCS patients (similar to the results in which women were also included). 

### 2.4. MicroRNA Expression Levels—Another Approach: Patients with Left Main and Proximal LAD 

The analysis between unstable angina patients together with stable patients with significant lesions in the LM and proximal LAD (“UA and LM/proxLAD group”) compared to chronic coronary syndrome patients with significant lesions in other places than the LM and proximal LAD (“CCS other group”) is another approach to plaque vulnerability. In this analysis, it is assumed that not only patients with clinical characteristics (i.e., unstable angina) but also patients with stable clinical phenotype and significant lesions in the left main or proximal left anterior descending are characterized with more vulnerable plaque phenotypes. It was demonstrated that patients from the first group (i.e., UA and LM or proximal LAD lesions) presented an increase not only in miR-10b-5p, but also in miR-92a-3p expression levels compared to CCS patients with lesions in other areas of coronary bed. The results and comparison between “UA and LM/proxLAD group” (n = 29 patients, 2 patients initially excluded from the main study due to significant lesions in left main and stable clinical phenotype [Figure 4] were now included) and the “CCS other group” (n = 25 patients) are showed in Table 4 and Figure 5, respectively. The differences in the expression level of miR-92a-3p and miR-10b-5p (in particular significant differences) in the aforementioned groups were presented in Figure 6 and Figure 7, respectively. 

## 3. Materials and Methods

### 3.1. Study microRNAs

The microRNA specific to mRNAs of atheroprotective proteins (IL10, MerTK, TGFβ1 and TGFβ3, KLF2, and KLF4) were selected according to the miRTarBase 7.0 and presented in Table 5 [22]. 

### 3.2. Study Population

Twenty patients were initially qualified as supposed unstable angina (UA) patients (15 consecutive patients hospitalized in the period of November 2022–July 2023 and 5 frozen plasma samples of patients hospitalized in the period June 2017—January 2018), and 50 patients were admitted for diagnostic evaluation of chronic coronary syndrome (CCS) were initially qualified as supposed chronic coronary syndrome. All the studied patients were admitted to the 1st Department of Cardiology, Central Clinical Hospital, University Clinical Centre, Medical University of Warsaw. Initially, 13 volunteers were also enrolled in the study. The study was approved by the Bioethics Committee of the Medical University of Warsaw (number KB/88/2022 and KB 54/A/2022).

### 3.3. Eligibility Criteria 

Patients with supposed chronic coronary syndrome were admitted to the hospital for coronary angiography. Indications for coronary angiography in this group were either the increased probability of CCS or the results of non-invasive tests (e.g., stress echocardiography, computed tomography coronary angiography—CTCA, treadmill test, single-photon emission computed tomography—SPECT) indicative for high-risk events, according to the European Society of Cardiology (ESC) guidelines [23]. 

Patients with supposed unstable angina were admitted to the hospital through the Emergency Department due to chest pain indicating unstable angina, according to corresponding ESC guidelines. These patients underwent coronary angiography and presented negative troponin tests [24]. 

Healthy volunteers presented with no history of any chronic disease, no history of surgical intervention or infection within the previous 3 months before blood taking, no artificial material in the body, and no signs of any disease in laboratory tests (i.e., complete blood count test and biochemical test, including lipid profile). Among 13 healthy volunteers (HV), 3 volunteers were diagnosed with dyslipidemia or anemia, and these volunteers were also excluded from the study. 

All the patients were eligible for the exclusion criteria (Table 6).

Patients initially qualified as supposed UA and CCS underwent coronary angiography (except for four patients with supposed CCS, the decision was made to withdraw from coronary angiography, and these patients were excluded from the study—more details in Figure 1). Among CCS patients, only those in whom coronary angiography showed at least one mild coronary artery stenosis (i.e., stenosis of ≥30%) and no significant lesion in the left main (LM) were enrolled in the study [27]. The UA patients met the same inclusion criteria, but the majority of UA patients presented significant coronary stenotic lesions (according to ESC guidelines for heart revascularization) and were qualified for coronary intervention (either percutaneous coronary intervention or coronary artery bypass grafting [28]. 

The final number of individuals enrolled in the study were n = 38 CCS patients, n = 14 UA patients, and n = 10 healthy volunteers (Figure 4). 

### 3.4. Blood Samples

In the CCS group, blood samples were taken in the morning, before the coronary angiography procedure, whereas in the UA group, blood samples were taken within the first 24 h after admission and more than 12 h after heparin administration to the patient. In the healthy volunteers, blood samples were taken in the morning, on fast. Blood samples in EDTA tubes were centrifuged at 1200× *g* for 10 min, separated plasma samples were aliquoted to microcentrifuge tubes, frozen on dry ice, and stored at −80 °C within 4 h from the blood draw. In every subject (including healthy volunteers), additional blood parameters were measured, i.e., complete blood count (with hemoglobin [Hgb] concentration), lipid profile (total cholesterol level, LDL-C, HDL-C, triglycerides), TSH, AspAT, AlAT, glycemia, creatinine (and eGFR) and troponin I (only in the UA group). 

### 3.5. Micro-RNA

The miRNA particles measured in the plasma samples were cel-miR-39-3p (a “spike-in” control), hsa miR-93-3p (abbrev. miR-93), and hsa-miR-191-5p (abbrev. miR-191) as reference microRNAs and 5 study miRNAs—hsa-miR-92a-3p, hsa-miR-10b-5p, hsa-miR-126-3p, hsa-miR-98-5p, and hsa-miR-29b-3b. Serum samples were thawed on ice and centrifuged at 20,000× *g* for 15 min at 4 °C. Clear supernatant (300 µL) was transferred to a new DNAse/RNAse free tube and mixed with a denaturing buffer (300 µL) and vortexed for 15 s. After 5 min of incubation, 30 fmol of 5′-phosphorylated synthetic cel-miR-39-3p (catalog no:10620310) was added to each sample as a “spike-in” control. The procedure of further microRNA isolation was consistent with manufacturer guidelines of mirVana™ miRNA Isolation Kit (AM1560, Thermo Fischer Scientific, Waltham, MA, USA). Then the RNA was reverse transcribed using TaqMan Advanced miRNA cDNA Synthesis Kit (Thermo Fisher Scientific). Libraries of cDNA were stored at −20 °C until further steps. The microRNA expression levels were assessed using PowerUp SYBR Green Master Mix (A25777 Thermo Fisher Scientific) according to the manufacturer’s protocol. The primers for each measured miRNA were purchased from the Institute of Biochemistry and Biophysics, Polish Academy of Sciences. Primers sequences were obtained from miRbase.org and were as follows: external control (“spike-in”) cel-miR-39-3p (TCACCGGGTGTAAATCAGCTTG), internal control miRNAs hsa-mir-191-5p (CAACGGAATCCCAAAAGCAGCTG) and hsa-mir-93-3p (ACTGCTGAGCTAGCACTTCCCG) as well as study miRNA particles—hsa-mir-92a-3p (TATTGCACTTGTCCCGGCCTGT), hsa-miR-10b-5p (TACCCTGTAGAACCGAATTTGTG), hsa-miR-126-3p (TCGTACCGTGAGTAATAATGCG), hsa-miR-98-5p (TGAGGTAGTAAGTTGTATTGTT), and hsa-miR-29b-3p (TAGCACCATTTGAAATCAGTGTT) All qPCRs were conducted in MicroAmp Fast Optical 96 Well Reaction Plates (Thermo Fisher Scientific) in total volume of 20 µL. All the reactions were executed on the Applied Biosystems ViiA7 Real-Time PCR System with ViiA^TM^ 7 Software Version 1.2.2. (Life Technologies, Thermo Fisher Scientific, Waltham, MA, USA. ViiA 7). All samples were assayed in triplicates. The obtained Ct (cycles of threshold) values for study miRNAs—hsa-mir-92a-3p, hsa-miR-10b-5p, hsa-miR-126-3p, hsa-miR-98-5p and hsa-miR-29b-3p as well as for internal control miRNAs—hsa-miR-191-5p and hsa-miR-93-3p were normalized by Ct of cel-miR-39-3p and were used to calculate relative expression using the 2-∆∆Ct method (a study microRNA referred to the mean of two reference microRNAs—miR-93 and miR-191). 

### 3.6. Statistical Analysis

The statistical analyses were done with the use of STATISTICA 13, StatSoft Inc.). Values are presented as mean and standard deviation (SD) for normally distributed variables and as median and interquartile range for non-normally distributed values. The normality tests were done by the use of the Shapiro-Wilk test. The Student’s *t*-test (for equal variances), Cochrane-Cox test (for non-equal variances), and ANOVA F-test were used for the comparison of normally distributed continuous variables, whereas Kruskall-Wallis test and Mann-Whitney U test—for non-normally-distributed-continuous variables. Categorical variables were compared by the use of χ2 test. The probability (*p*) values < 0.05 were considered statistically significant. The effect-size r coefficient was calculated based on Cohen’s d, by the use of the Effect Size Calculator (Dr. Lee E. Beckers, University of Colorado).

## 4. Discussion

Our study revealed that patients with unstable angina presented significantly increased expression levels of miR-10b compared to patients with chronic coronary syndrome. This correlation also remained when only male populations were compared. UA population was slightly younger than CCS population, there were more active smokers, HDL levels were lower (without significant changes in other parameters of lipid profile), and a majority of them (85.7%) presented indications for the invasive treatment (compared to chronic coronary syndrome group in which 52.6% of patients presented such indications). Although the HV group was younger than other groups, there were 60% of women, and BMI was significantly lower than in the UA group, there were no statistically significant differences between the HV group and any other group in every investigated microRNA. There were studies that aimed to find the expression profile of different microRNAs in patients with unstable angina [29,30]. Some studies concentrated on finding certain miRNA as a biomarker of acute coronary syndrome, distinguishing it from non-coronary pain in the A&E Department [31,32,33]. Other studies concentrated on differences between unstable angina and other manifestations of coronary artery disease (like stable angina, currently defined by the term “chronic coronary syndrome” or myocardial infarction with dynamic troponin elevation) [30,34,35,36]. Navickas et al. prepared a meta-analysis presenting promising microRNAs as potential diagnostic or prognostic biomarkers across different cardiovascular disease progression stages [37]. There were also studies aiming to elucidate mechanisms of atherosclerotic plaque progression with the involvement of miRNA [38]. The study most similar to ours was conducted by Cui et al., in which elderly people with diagnosed stable angina and unstable angina were compared for miRNA expression [39]. Initially, the microarray study was conducted with 10 patients from both groups, and then qRT-PCR was conducted on 30 stable angina and 30 unstable angina patients, demonstrating slight differences in miR-1202, miR-1207, and miR-1225 and a marked difference in miR-3162 expression levels. Yet, in this study, there were no exclusion criteria like a history of myocardial infarction, history of stent implantation, artificial element in the body, history of stroke, and diabetes, which were applied in our study. The strenuous exclusion criteria, like a history of previous coronary intervention or presence of aortic aneurysm, were applied in our study to minimize the influence of processes like epithelialization, extracellular matrix remodeling, and vascular remodeling after injury. For instance, the presence of aortic aneurysm is correlated with changes in microRNAs expression (especially microRNAs affecting extracellular matrix remodeling) [40]. Among studies aimed to investigate mechanisms of plaque progression, there are studies concentrating particularly on the mechanism of plaque destabilization. Huang, in his review, summarizes the information on microRNAs promoting plaque destabilization and rupture and their target mechanisms within the plaque [41]. In this context, our study aims to investigate differences in the expression level of five microRNA, where all of them target different atheroprotective proteins regulating different processes and stages of atherosclerotic plaque progression. This approach assumption also aimed to investigate the importance of atheroprotective factors as targets regulated by specific microRNA particles and potential relations between these networks in different clinical contexts of coronary artery disease. The clinical characteristic of instability reflects the vulnerable character of the plaque, yet, in chronic coronary syndrome, a clinical presentation of a stable atherosclerotic plaque doesn’t mean that there are no elements of instability in atherosclerotic plaques observed in this group of patients. A vulnerable phenotype of plaque might be assessed by intracoronary imaging techniques like intravascular ultrasound—virtual histology (IVUS-VH) or optical coherence tomography (OCT), but these techniques are not used routinely in clinic [42]. Nevertheless, it has been shown that the plaques localized in the left main and proximal left coronary descending present a more unstable phenotype than plaques localized in other regions [43,44,45]. Therefore, the patients enrolled in this study (UA and CCS) were divided in another way. To UA patients, there were also added patients with LM and proximal LAD lesions (“UA and LM/proxLAD group”; n = 29 patients), and this group was compared with CCS patients with lesions in other regions of coronary bed than LM and proximal LAD (“CCS other group”; n = 25 patients) with the assumption that the former group presents plaques with more vulnerable phenotype. This subanalysis demonstrated that “UA and LM/proxLAD group” presented not only miR-10b, but also miR-92a significantly elevated, compared to “CCS other group”. It is necessary to point out that both of these microRNAs are involved in endothelial regulation, inhibiting endothelial protective factors KLF4 and KLF2, respectively [46,47]. Last but not least, microRNAs also regulate physiological and pathophysiological processes in peripheral arteries. The processes of peripheral arterial disease pathogenesis resemble, albeit not in 100%, the process of atherogenesis in coronary arteries [48]. Therefore, this process might influence the microRNA profile in such patients. In our study, there were no significant differences in PAD patients between the study groups.

## 5. Conclusions

The results of our study might suggest that pathways involved in the regulation of endothelial factors might be crucial in the progression and destabilization of atherosclerotic plaque that manifests clinically as an unstable angina. However, there are limitations of our study, including a low number of patients, younger age of UA patients, some differences in baseline characteristics (especially younger age of healthy volunteers), and a relatively small effect size (r < 0.4 in all statistically significant results). We aimed to select the homogenous groups of patients free from other diseases that would be likely to confound the analysis (like previous myocardial infarction, previous stroke, diabetes, inflammation, and active cancer). 

## Figures and Tables

**Figure 1 ijms-25-10621-f001:**
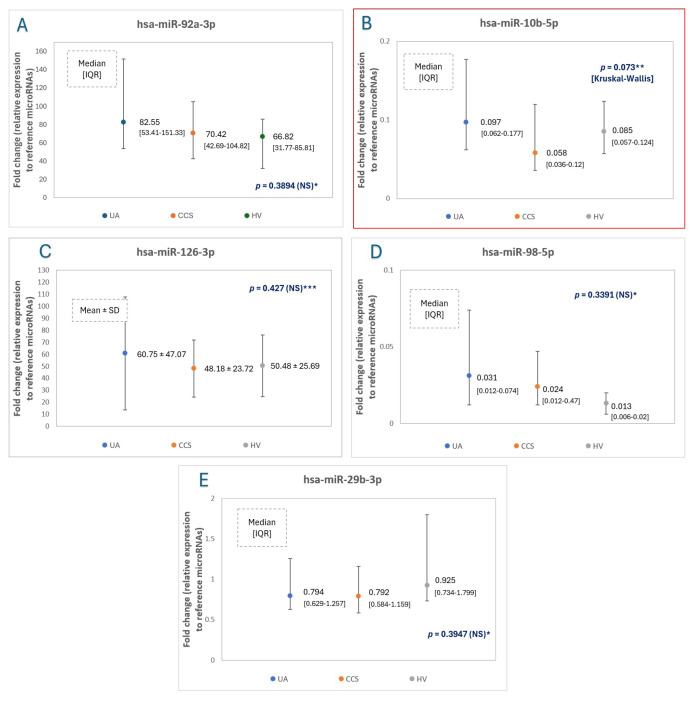
The relative expression levels of study microRNAs ((**A**)—hsa-miR-92a-3p; (**B**)—hsa-miR-10b-5p, (**C**)—hsa-miR-126-3p, (**D**)—hsa-miR-98-5p, (**E**)—hsa-miR-29b-3p) to reference microRNAs (mean of miR-93 and miR-191 expression levels) in unstable angina (UA) patients comparing to CCS (chronic coronary syndrome—stable angina) patients and healthy volunteers (HV). *—Normality rejected, ANOVA Kruskall-Wallis test applied, **—value for ANOVA Kruskall-Wallis test, pair comparison, U-Mann Whitney tests were applied, and results were presented in Figure 2, ***—normality non-rejected, equal variances in Levene test, F-ANOVA test applied. *p* < 0.05 is considered statistically significant.

**Figure 2 ijms-25-10621-f002:**
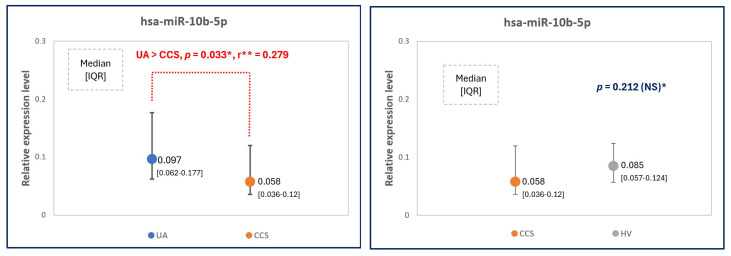
The relative expression levels of hsa-miR-10b-5p to reference microRNAs (mean of miR-93 and miR-191 expression levels) in unstable angina (UA) patients compared to CCS (chronic coronary syndrome—stable angina) patients and healthy volunteers (HV). *—Normality rejected, U-Mann Whitney tests were applied, *p* < 0.05 considered as statistically significant **—effect size calculated by Cohen’s d.

**Figure 3 ijms-25-10621-f003:**
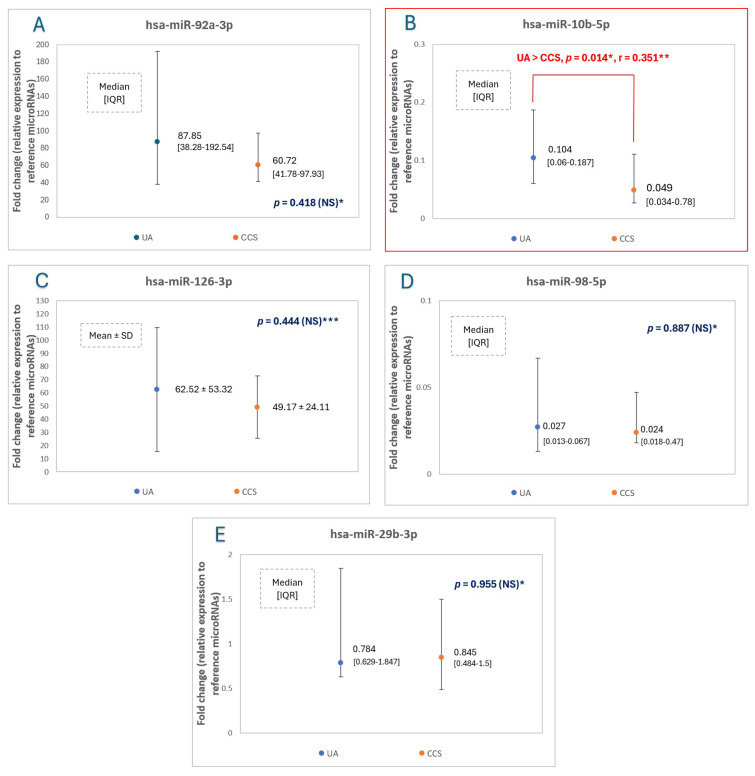
The relative expression levels of study microRNAs ((**A**)—miR-92a; (**B**)—miR-10b, (**C**)—miR-126, (**D**)—miR-98, (**E**)—miR-29b) to reference miRNAs (the mean of miR-93 and miR-191 expression levels) in unstable angina (UA) comparing to CCS (chronic coronary syndrome—stable angina) male patients. *—Normality rejected, U-Mann Whitney Test was applied, **—effect size, assessed by Cohen’s d; ***—normal distribution, non-equal variances in Levene test, Cochrane-Cox test applied. *p* < 0.05 is considered statistically significant.

**Figure 4 ijms-25-10621-f004:**
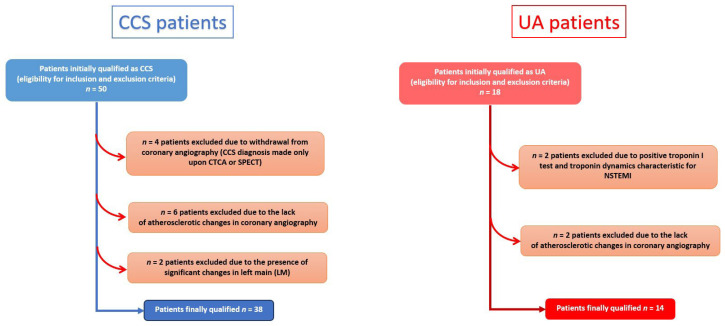
Study flow chart. CCS—chronic coronary syndrome, UA—unstable angina.

**Figure 5 ijms-25-10621-f005:**
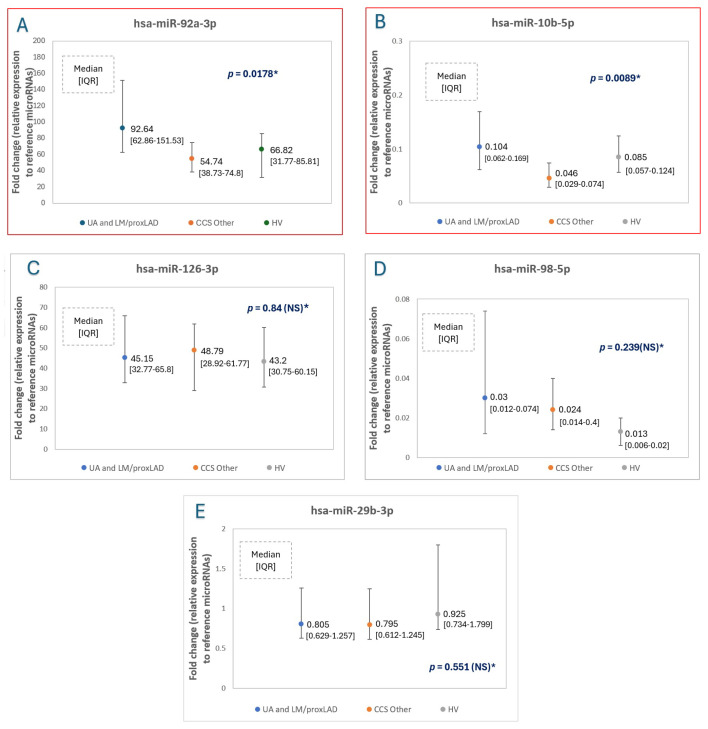
The relative expression levels of study microRNAs ((**A**)—hsa-miR-92a-3p; (**B**)—hsa-miR-10b-5p, (**C**)—hsa-miR-126-3p, (**D**)—hsa-miR-98-5p, (**E**)—hsa-miR-29b-3p) to reference microRNAs (mean of miR-93 and miR-191 expression levels) in unstable angina patients and patients with left main or proximal left anterior descending (“UA and LM/proxLAD”) group, CCS patients with atherosclerotic lesions in other places in coronary artery bed (“CCS other” group) and healthy volunteers (HV). *—Normality rejected, ANOVA Kruskall-Wallis test was applied, *p* < 0.05 is considered statistically significant.

**Figure 6 ijms-25-10621-f006:**
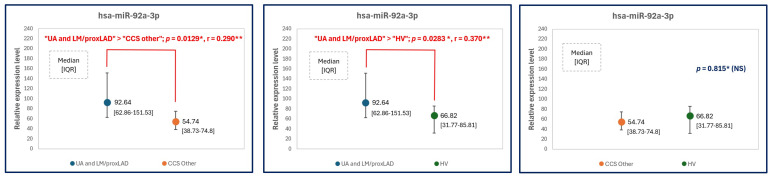
The relative expression levels of hsa-miR-92a-3p to reference microRNAs (the mean of miR-93 and miR-191 expression levels) in unstable angina patients and patients with left main or proximal left anterior descending (“UA and LM/proxLAD”) group, CCS patients with atherosclerotic lesions in other places in coronary artery bed (“CCS other” group) and healthy volunteers (HV). *—Normality rejected, U-Mann Whitney tests were applied, *p* < 0.05 considered as statistically significant, **—effect size assessed by Cohen’s d.

**Figure 7 ijms-25-10621-f007:**
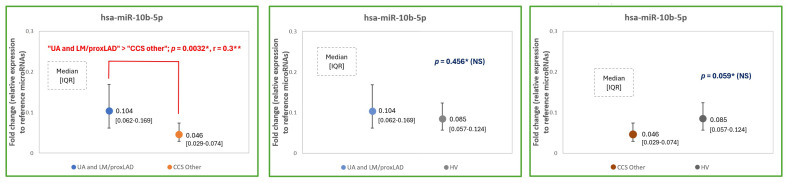
The expression levels of hsa-miR-10b-5p to reference microRNAs (mean of miR-93 and miR-191 expression levels) in unstable angina patients and patients with left main or proximal left anterior descending (“UA and LM/proxLAD”) group, CCS patients with atherosclerotic lesions in other places in coronary artery bed (“CCS other” group) and healthy volunteers (HV). *—Normality rejected, ANOVA Kruskall-Wallis test was applied, *p* < 0.05 is considered statistically significant, **—effect size assessed by Cohen’s d.

**Table 1 ijms-25-10621-t001:** Baseline characteristics for unstable angina (UA), chronic coronary syndrome (CCS) patients, and healthy volunteers (HV). * *p* value <0.05 considered as statistically significant; values with normal distribution and homogenous variation presented as “mean ± SD (standard deviation)”, F-ANOVA statistics test (with Scheffè post-hoc analysis) and Student’s *t*-test (for APTT) applied; values with rejected distribution normality or unequal variances presented as “median [interquartile range]”, ANOVA Kruskal-Wallis test and Mann Whitney U-tests applied; ** quantitative values presented as “number (percentage in group)”, χ^2^ test (with Yates correction) applied, ***—indications for invasive treatment means that the patients were qualified either for PCI or for CABG or for conservative treatment but not due to the absence of a significant plaque but due to clinical contraindications—old age, comorbidities, inaccessibility of the atherosclerotic lesion for invasive treatment.

	UA Patients (n = 14)	CCS Patients (n = 38)	HV (n = 10)	*p* Significance *
Age [years]	62 ± 8	71 ± 11	39 ± 10	Significant between − HV and UA (*p* = 0.000004) − HV and CCS (*p* < 0.000001) − CCS and UA (*p* = 0.03)
Sex (woman) **	3 (21.43%)	18 (47.4%)	6 (60%)	NS (*p* = 0.13)
Hypertension **	11 (84.6%)	30 (81.1%)	NA	NS (*p* = 0.78)
Hyperlipidemia **	9 (69.2%)	26 (68.4%)	NA	NS (*p* = 0.99)
Active smoker **	7 (53.9%)	7 (18.4%)	NA	Significant (*p* = 0.013)
Previous smoker **	4 (30.8%)	9 (23.7%)	NA	NS (*p* = 0.61)
CAD or PAD **	5 (35.7%)	6 (15.8%)	NA	NS (*p* = 0.12)
Baseline statin use **	5 (30.8%)	16 (42.1%)	NA	NS (*p* = 0.47)
Baseline arrythmia **	0 (0%)	6 (15.8%)	NA	NS (*p* = 0.13)
BMI [kg/m^2^]	29.5 ± 4.9	25.8 ± 4.0	24.0 ± 3.6	− Significant between UA and HV (*p* = 0.02) − NS between UA and CCS (*p* = 0.06) − NS between HV-CCS (*p* = 0.56)
TC [mg/dL]	196.5 [129–224]	150.5 [130–190]	183.5 [173–203]	NS (*p* = 0.18)
LDL [mg/dL]	93.5 [62–132.5]	78 [61–108]	97 [89–117]	NS (*p* = 0.25)
HDL [mg/dL]	37.8 ± 3.9	51.3 ± 11.6	65.1 ± 10.5	Significant between − UA and CCS (*p* = 0.00018) − UA and HV (*p* = 0.000011) − CCS and HV (*p* = 0.003)
TG [mg/dL]	142.5 [94–187]	103 [80–138]	76.5 [67–110]	− Significant between UA and HV (*p* = 0.007), − NS between UA and CCS (*p* = 0.07); − NS between CCS and HV (*p* = 0.12)
eGFR [mL/min/1.73 m^2^]	74.1 ± 17.5	66.1 ± 18.7	99.5 [90–113]	− Significant between UA and HV (*p* = 0.02) − CCS and HV (*p* = 0.0005) − NS between CCS and UA
Hgb [g/dL]	14.9 ± 1.7	13.7 ± 1.4	13.9 ± 1.5	− Significant between UA and CCS (*p* = 0.04) − NS between CCS and HV (*p* = 0.9) − NS between UA and HV (*p* = 0.27)
PLT [number]	243 ± 64	228 ± 71	241 ± 61	NS (*p* = 0.73)
TSH [μIU/mL]	1.41 [0.929–2.39]	1.39 [0.848–2.42]	1.34 [0.641–2.06]	NS (*p* = 0.80)
AspAT [U/L]	24 [19.5–26.5]	25 [21–30]	NA	NS (*p* = 0.30)
AlAT [U/L]	35 [24–52]	24 [17–29]	NA	Significant (*p* = 0.008)
APTT [s]	30.1 ± 3.6	29.4 ± 3.6	NA	NS (*p* = 0.57)
**NEXT STEP AFTER CORONARY ANGIOGRAPHY**				
Indications for invasive treatment ***	12 (85.7%)	20 (52.6%)	NA	Significant (*p* = 0.03)

**Table 2 ijms-25-10621-t002:** The study microRNA expression level values in patients with unstable angina (UA), chronic coronary syndrome (CCS), and healthy volunteers (HV). *—fold changes relative to reference microRNAs (mean of miR-93 and miR-191 expression levels), values expressed by median [IQR, i.e., interquartile range] for non-normally distributed miRNAs values and by mean [SD, i.e., standard deviation] for normally distributed values and equal variances in F-ANOVA test (miR-126); ** *p*-significance, <0.05 considered as statistically significant, ***—difference between miR-10b expression levels in Kruskal-Wallis test, in Mann-Whitney U-test difference between UA and CCS group was statistically significant (presented on Figure 3).

microRNAs	Fold Change in UA Group (n = 14) *	Fold Change in CCS Group (n = 38) *	Fold Change in HV Group (n = 10) *	*p* **
Hsa-miR-92a-3p	82.55 [53.41–151.53]	70.42 [42.69–109.82]	66.82 [31.77–85.81]	NS (0.389)
Hsa-miR-10b-5p	0.097 [0.062–0.177]	0.058 [0.036–0.120]	0.085 [0.057–0.124]	0.073 (Kruskal-Wallis test) ***
Hsa-miR-126-3p	60.75 ± 47.07	48.18 ± 23.72	50.48 ± 25.69	NS (0.427)
Hsa-miR-98-5p	0.031 [0.012–0.074]	0.024 [0.012–0.047]	0.013 [0.006–0.020]	NS (0.339)
Hsa-miR-29b-3p	0.794 [0.629–1.257]	0.792 [0.584–1.159]	0.925 [0.734–1.799]	NS (0.395)

**Table 3 ijms-25-10621-t003:** The study microRNA expression level values in male patients with unstable angina (UA) and chronic coronary syndrome (CCS). *—fold changes relative to reference microRNAs (mean of miR-93 and miR-191 expression levels), values expressed by median [IQR, i.e., interquartile range] for non-normally distributed miRNAs values and by mean [SD, i.e., standard deviation] for normally distributed values. ** *p*-significance, <0.05 considered as statistically significant, for hsa-miR-126-3p Cochrane-Cox test was applied (normal distribution, non-equal variances in Levene test), for other microRNAs—U-Mann-Whitney tests were applied.

microRNAs	Fold Change in UA Males (n = 11) *	Fold Change in CCS Males (n = 22) *	*p* **
Hsa-miR-92a-3p	87.85 [38.28–192.54]	60.72 [41.78–97.93]	NS (0.418)
Hsa-miR-10b-5p	0.104 [0.06–0.187]	0.049 [0.034–0.078]	0.014 **
Hsa-miR-126-3p	62.52 ± 53.32	49.17 ± 24.11	NS (0.444)
Hsa-miR-98-5p	0.027 [0.013–0.067]	0.024 [0.018–0.047]	NS (0.887)
Hsa-miR-29b-3p	0.784 [0.629–1.847]	0.845 [0.484–1.5]	NS (0.955)

**Table 4 ijms-25-10621-t004:** The study microRNA relative expression level values to reference microRNAs (mean of miR-93 and miR-191 expression levels) in patients with significant changes in unstable angina patients and patients with left main or proximal left anterior descending (“UA and LM/proxLAD”) group, CCS patients with atherosclerotic lesions in other places in coronary artery bed (“CCS other” group) and healthy volunteers (HV). *—values expressed by median [IQR, i.e., interquartile range] for miRNA expression levels due to rejected normality in Shapiro-Wilk test, ** *p*-significance, <0.05 considered as statistically significant, ANOVA Kruskal-Wallis test applied, ***—intergroup analysis presented on Figure 3.

microRNAs	Fold Change in ”UA and LM/proxLAD” (n = 29) *	Fold Change in ”CCS Other” (n = 25) *	Fold Change in HV (n = 10) *	*p* **
Hsa-miR-92a-3p	92.64 [62.86–151.53]	54.74 [38.73–74.80]	66.82 [31.77–85.81]	0.0178 ***
Hsa-miR-10b-5p	0.104 [0.062–0.169]	0.046 [0.029–0.074]	0.085 [0.057–0.124]	0.0089 ***
Hsa-miR-126-3p	45.15 [32.77–65.8]	48.79 [28.92–61.77]	43.2 [30.75–60.15]	NS (0.84)
Hsa-miR-98-5p	0.03 [0.012–0.074]	0.024 [0.014–0.04]	0.013 [0.006–0.020]	NS (0.239)
Hsa-miR-29b-3p	0.805 [0.629–1.257]	0.795 [0.612–1.245]	0.925 [0.734–1.799]	NS (0.551)

**Table 5 ijms-25-10621-t005:** The microRNAs and their target mRNAs of atheroprotective proteins.

mRNA of the Protein	Corresponding Downregulating microRNA
IL10	Hsa-miR-98-5p (abbrev. miR-98)
MerTK	Hsa-miR-126-3p (abbrev. miR-126)
TGFβ1 and TGFβ3	Hsa-miR-29b-3p (abbrev. miR-29b)
KLF2	Hsa-miR-92a-3p (abbrev. miR-92a)
KLF4	Hsa-miR-10b-5p (abbrev. miR-10b)

**Table 6 ijms-25-10621-t006:** The exclusion criteria for the study population. *—according to KDIGO guidelines [25], **—according to ADA guidelines [26].

Exclusion Criteria for Chronic Coronary Syndrome (CCS) Patients
A history of malignant neoplastic disease and active malignant neoplastic disease Active autoimmune or rheumatic disease Surgery or invasive intervention within a period of 6 months before sample-taking Chronic kidney disease (G3b, G4, G5 according to KDIGO 2012) and eGFR level (CKD-EPI) < 45 mL/min/1.73 m^2^ * Diabetes mellitus (according to ADA—American Diabetes Association—Standards of Medical Care in Diabetes—2018) ** Congestive Heart Failure (NYHA III/IV) Diagnosis of active infection or history of infection within the period of 3 months before sample taking. A history of previous acute coronary syndrome A history of cerebrovascular stroke or TIA (transient ischemic attack) A history of previous percutaneous coronary intervention with stent implantation. A history of any cardiosurgical procedure A history of aortic or heart wall aneurysm Any artificial element in the body (prosthesis, pacemaker system, vascular draft, embolization coils)
Exclusion criteria for unstable angina (UA) patients
*Same criteria like for CCS patients* AND Positive troponin test, indicative of non-ST-segment elevation myocardial infarction (NSTEMI)

## Data Availability

The anonymized data presented in this study are available on request from the corresponding author (M.K.) due to privacy restrictions.

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
