# Peer review of "MicroRNA Inhibiting Atheroprotective Proteins in Patients with Unstable Angina Comparing to Chronic Coronary Syndrome"

_ijms, 2024, doi:10.3390/ijms251910621_

Round 1

Reviewer 1 Report

Comments and Suggestions for Authors

 Manuscript is interesting, but I have more questions, added below.

Is the mechanism of action of MicroRNA defined in relation to other arteries? If there is, add the data briefly  to the discussion. 

Why the perspective of surgery was the exclusion criterion?

Why a history of previous percutaneous coronary intervention or coronary bypass were exclusion criteria? If the patient had undergone stenting or bypass surgery and had atherosclerotic plaque progression in another artery, why was this condition considered an exclusion criteria?

Why a history  of aortic or heart wall aneurysm was exclusion criterion?

It seems that the blood was taken once.  Were the patients under follow-up and for what period?

Author Response

REVIEWER 1:

Comments and Suggestions for Authors: Manuscript is interesting, but I have more questions, added below.
Is the mechanism of action of MicroRNA defined in relation to other arteries? If there is, add the data briefly  to the discussion. 
Why the perspective of surgery was the exclusion criterion?
Why a history of previous percutaneous coronary intervention or coronary bypass were exclusion criteria? If the patient had undergone stenting or bypass surgery and had atherosclerotic plaque progression in another artery, why was this condition considered an exclusion criteria?
Why a history  of aortic or heart wall aneurysm was exclusion criterion?
It seems that the blood was taken once.  Were the patients under follow-up and for what period?

RESPONSE TO REVIEWER 1: 
We are very grateful for Your assessment and Your insightful questions. Responding briefly on these questions: 
- yes, microRNA particles were also investigated in other arteries, we have added the information and corresponding reference regarding this aspect to the discussion. 
- the history of by-pass surgery and PTCA, as well as the presence of aortic aneurysm were exclusion criteria because both PTCA and CABG alter the coronary vessels and induce a plethora of processes involved in vascular remodelling which might affect the physiology of such vessel in future. In reference to aortic aneurysm - within the aneurysm the process of extracellular matrix remodelling occurs and aortic aneurysms progression (at least in mice) mimics the processes of plaque destabilization. 
- the perspective of surgery was not an exclusion criterium, but the history of surgery in the period of 6 months before sample taking - due to potential interference with processes of vascular remodelling.
- Thank You for the last comment, but our study aimed to distinguish two pathophysiological situations (UA and CCS), therefore follow-up was not included in this study. 

Reviewer 2 Report

Comments and Suggestions for Authors

This paper by Kowara Michal et al aims to investigate the role of specific microRNAs in inhibiting atheroprotective proteins, contributing to atherosclerotic plaque vulnerability in patients with unstable angina (UA) compared to those with chronic coronary syndrome. The study identifies increased levels of miR-10b and miR-92a in UA patients, suggesting their involvement in plaque destabilization. The main strength of the study lies in its focused analysis of miRNAs as potential biomarkers for plaque vulnerability. However, its weaknesses include a small sample size and demographic differences between study groups, which may limit the generalizability of the findings.

I have two comments:

1) While the introduction provides a good overview, it could benefit from a more detailed discussion on the broader context of microRNAs in cardiovascular diseases and the research gap that this study aims to fill. For example, the introduction could include more references to previous studies that link specific microRNAs to atherosclerosis and plaque stability and an explanation why the selected microRNAs were chosen over others.

2) The results could be presented more effectively with clearer or additional graphs or charts that make it easier to compare the groups and understand the differences in microRNA expression levels. For example, more detailed labeling or annotated graphs could highlight the significant findings better. Further, the interpretation of statistical data could be more detailed. For example, discussing the effect sizes or providing confidence intervals alongside p-values would give a better sense of the magnitude and reliability of the findings.

Comments on the Quality of English Language

The English is generally understandable, but it contains grammatical errors.

Author Response

Reviewer 2        

Major

Comments and Suggestions for Authors

This paper by Kowara Michal et al aims to investigate the role of specific microRNAs in inhibiting atheroprotective proteins, contributing to atherosclerotic plaque vulnerability in patients with unstable angina (UA) compared to those with chronic coronary syndrome. The study identifies increased levels of miR-10b and miR-92a in UA patients, suggesting their involvement in plaque destabilization. The main strength of the study lies in its focused analysis of miRNAs as potential biomarkers for plaque vulnerability. However, its weaknesses include a small sample size and demographic differences between study groups, which may limit the generalizability of the findings.

I have two comments:

1) While the introduction provides a good overview, it could benefit from a more detailed discussion on the broader context of microRNAs in cardiovascular diseases and the research gap that this study aims to fill. For example, the introduction could include more references to previous studies that link specific microRNAs to atherosclerosis and plaque stability and an explanation why the selected microRNAs were chosen over others.

2) The results could be presented more effectively with clearer or additional graphs or charts that make it easier to compare the groups and understand the differences in microRNA expression levels. For example, more detailed labeling or annotated graphs could highlight the significant findings better. Further, the interpretation of statistical data could be more detailed. For example, discussing the effect sizes or providing confidence intervals alongside p-values would give a better sense of the magnitude and reliability of the findings.

RESPONSE TO REVIEWER 2: 

We are very grateful for these insightful comments. First of all, we performed additional statistical analysis including r-size effect (Cohen's d method, applied for statistically significant differences). The graphs were also modified to improve their legibility, additional 3 figures has been added. 
Second, we improved the discussion and introduction in which we aimed to present the context of our study, according to Your suggestion. 

Reviewer 3 Report

Comments and Suggestions for Authors

Manuscript  well written and interesting.

Is the mechanism of action of MicroRNA defined in relation to other arteries? If there is, add the data briefly  to the discussion

Author Response

Reviewer 3        

Minor

Comments and Suggestions for Authors

Manuscript  well written and interesting.

Is the mechanism of action of MicroRNA defined in relation to other arteries? If there is, add the data briefly  to the discussion

RESPONSE TO REVIEWER 3: 
We are very grateful for Your Review, we appreciate Your evaluation and suggestion, yes, we have added this aspect into the discussion. 

Round 2

Reviewer 2 Report

Comments and Suggestions for Authors

Dear Authors!

Thank you for the extensive revisions, which have significantly improved the quality of your paper. The changes are satisfactory to me, and I now believe it is suitable for publication.

Author Response

We are grateful for Your review and comments, please find our response in an attached file. 
